# M³Stroke: Multi-Modal Mobile AI for Emergency Triage of Mild to Moderate Acute Strokes

Tongan Cai*†, Kelvin Wong◇, James Z. Wang*†, Sharon Huang†, Xiaohui Yu◇, John J. Volpi‡, Stephen T.C. Wong*◇

*Abstract*—**Over 22% of ischemic stroke patients are overlooked during triage in the emergency departments, particularly those with mild or moderate stroke which resembles stroke mimics in symptoms. While pronounced neurological conditions can be captured with existing AI solutions, identifying stroke patients with minor symptoms remains under-explored due to data scarcity, noise complexity, and feature subtlety. We propose M³Stroke, a MultiModal Mobile AI tool, to enhance the accuracy and efficiency of stroke triage for these patients. As the first stroke screening tool to integrate novel audio-visual multimodal AI into efficient mobile computing, M³Stroke runs seamlessly on common iOS devices and significantly outperforms prior methods. Trained and evaluated on a dataset of 269 patients suspected of stroke (191 stroke/78 non-stroke), M³Stroke model achieves 80.85% accuracy, 60.00% specificity, and 90.63% sensitivity, demonstrating 14.29% gain in specificity and 20.44% higher sensitivity compared with traditional stroke triage methods. The tool's performance, robustness, and fairness across diverse demographics confirm its potential to improve ER triage, aiding tele-stroke detection and self-diagnosis, and enhancing life quality for elderly patients.**

*Index Terms*—**Stroke, Artificial Intelligence, Computer Aided Diagnosis, Mobile Computing**

## I. INTRODUCTION

**S**TROKE stands as a significant global health crisis. In the United States, stroke is ranked as the fifth leading cause of death and is also a leading cause of long-term disability, with an estimated 795,000 people experiencing a stroke each year [1]. In cases where an acute stroke is suspected based on computerized tomography (CT), neurologists will utilize tools such as the NIH Stroke Scale to quantify neurological impairment and employ advanced imaging to confirm potential large vessel occlusions. The diffusion-weighted Magnetic Resonance Imaging (MRI) serves as the benchmark for stroke confirmation when accessible, but in its absence, the neurologist's judgment is often the primary diagnostic standard.

A recent report issued by the Agency for Healthcare Research and Quality under the U.S. Department of Health and Human Services estimated that stroke misdiagnosis is common in emergency rooms across the country [2], leading to under-treatment and over-treatment, increasing the risk of long-term mental impairments and physical disabilities, eventually driving up healthcare costs. Mild to moderate acute strokes often present with subtle neurological symptoms, which can be mistakenly attributed to less critical conditions, leading to oversight by both patients and healthcare professionals [3], compromising the quality of care and patient outcomes.

The recent advances in artificial intelligence (AI) and machine learning (ML) have catalyzed a transformative shift in medical diagnostics. However, the exploration of radiation-free, cost-effective screening methods remains limited, with existing research on AI-based acute ischemic stroke detection predominantly focused on severe cases [4]–[6]. There is a notable gap in AI research regarding mild to moderate strokes and stroke mimics due to the scarcity of patient data, the subtle manifestations of symptoms, and the need for advanced, efficient methods capable of modeling temporal dynamics.

To enhance the triage process in hospitals for acute stroke cases that present with mild to moderate symptoms, this study introduces a non-invasive, high-accuracy, AI-powered mobile framework, M³Stroke. This innovative tool employs an AI algorithm that analyzes patients' facial expressions and speech patterns to estimate the likelihood of an acute stroke, precisely distinguishing real strokes from stroke mimics.

M³Stroke is fully deployed on iOS devices (iPad/iPhone) as an offline application, providing a swift and efficient evaluation process—including speech task recording, data pre-processing, and inference within three minutes. It is distinct from prior mobile applications for pre-hospital stroke assessment [7]–[9] as the first unified mobile tool to perform multi-media data mobile collection and on-board AI-based patient screening. The M³Stroke framework's core model (M³Encoder) features a novel audio-visual multimodal encoder that introduces the structure state-space sequence representation (S4 [10]) to learn sequence dependencies, integrates the latest state-of-the-art (SoTA) audio transformer (AST [11]) to capture sparser audio spectrum features, and incorporates a multi-scale (low- and high-resolution), multi-level (frame- and clip-level) feature fusion scheme to preserve and utilize the features effectively.

M³Stroke achieves significant performance improvements over human triage clinicians and strong AI baselines. Built and evaluated on our collected comprehensive dataset with 269 patients suspected of mild acute stroke, the M³Stroke model outperformed standard triage nurse assessments with a 16.82% increase in accuracy, a 20.44% increase in sensitivity, and a 14.29% increase in specificity for mild stroke cases. Compared with the prior-best mild stroke screening model,

⋆ Corresponding authors; Contact: cta@psu.edu, jwang@ist.psu.edu, stwong@houstonmethodist.org

† College of Information Sciences and Technology, The Pennsylvania State University, University Park, PA 16802, USA.

◇ The T.T. and W.F. Chao Center for BRAIN & Systems Medicine and Bioengineering, Houston Methodist Hospital, Houston, TX 77030, USA.

‡ Eddy Scurlock Comprehensive Stroke Center, Houston Methodist Hospital, Houston, TX 77030, USA.

*DeepStroke* [12] and the latest SoTA model for facial video understanding, MARLIN [13], the M³Stroke model achieves 10%-20% better performance metrics. The model is further verified for fairness across sexes, ethnicities, and races.

Our **contributions** in applying **M**ulti-**M**odal **M**obile AI to advance hospital triage for stroke patients are multifaceted, each significantly pushing the frontiers of stroke care.

- *Innovative multimodal AI:* We have developed an AI-assisted tool M³Stroke, based on a brand-new audio-visual multimodal framework that significantly enhances the precision of detecting mild to moderate acute strokes during hospital triage, setting a new benchmark in diagnostic accuracy. Extensive testing on a curated, comprehensive dataset from emergency room patients–comprising both authentic mild stroke cases and stroke mimics–affirms the superior performance and considerable potential clinical impact, positioning M³Stroke as the state of the art in stroke triage.
- *Advancing clinical usability:* The mobile deployment of M³Stroke is a critical stride towards its practical clinical application. This not only facilitates ease of use but also paves the way for forthcoming clinical trials, and elevates hospital triage for stroke and other neurological conditions by offering a transformative tool for medical professionals.
- *Extending stroke care beyond hospitals:* The versatility of M³Stroke extends beyond the confines of traditional hospital settings into other medical practices, including emergency medical transportation, rural medical clinics, nursing homes, tele-stroke services, and patient self-monitoring. This expansion will allow cutting-edge diagnostics to be accessible in diverse healthcare environments, ultimately contributing to a universal uplift in stroke patient care and outcomes.

## II. RELATED WORKS

Computerized detection of neurological disorders with imaging results has seen decades of effort [14], [15]. More recently, multimodal AI incorporating information across different modalities has gained traction, and the information fusion of electronic health records and medical images has demonstrated success in this domain [16], [17]. However, incorporating both facial and speech data for stroke diagnosis, a key aspect of triage physicians' assessment, has received minimal attention in previous studies due to challenges of (1) data scarcity, (2) noise complexity, and (3) feature subtlety.

Existing media datasets for neurological diseases [18]–[20] are limited in size, comprising synthetic data from healthy subjects, or lacking sufficient representation across key demographic variables. In clinical data collection, many face-related studies constrain the patient's head movement [21]–[23] to circumvent face alignment challenges. Speech-related studies [24]–[26] typically conduct recordings in quiet laboratories. These controlled environments do not reflect the diverse and noisy real-world conditions, and frameworks built on lab-collected, noise-free data eventually fail to generalize on clinical scenarios. The utilization of image-based facial asymmetry and key-point analysis [27] has been effective in identifying pronounced facial deficits (macro motions) but falls

short in capturing the nuanced spatiotemporal dynamics or addressing the inherent asymmetry in facial structures, both of which are essential for detecting minor stroke symptoms. Existing video-based methods, including *DeepStroke* [12], struggle to effectively model patient video data that exhibit variable temporal dynamics. This limitation extends to SoTA video-based methods like VideoMAE [28].

These challenges create significant barriers to the development of reliable patient self-assessment tools, as well as to the practical clinical application of existing methods.

## III. METHODS

### A. Data Collection

To support the study[1] of AI-assisted triage for mild to moderate acute strokes, we recruited patients admitted to the Eddy Scurlock Comprehensive Stroke Center at Houston Methodist Hospital in the Texas Medical Center. We emulate the triage personnel's perspective of view and video record the patients' front faces along with audio using a mobile phone. Participants are performing the picture description tasks while their facial video and audio are being recorded: repeating the sentence "It is nice to see people from my hometown" and describing a "Cookie Theft" picture.

*1) In-/ex-clusion criteria:* Our study meticulously crafted inclusion/exclusion criteria to concentrate on nuanced stroke diagnoses while safeguarding emergency patients from risk: we included individuals who sought or had sought ER care under suspicion of acute stroke yet did not exhibit pronounced stroke-related impairments or critical conditions. The final diagnoses, substantiated via diffusion-weighted MRI scans, determined the presence, type, location, and severity of any cerebral lesions, confirming or ruling out stroke occurrences.

Our study spanned from February 2019 to Oct 2022, with a diverse cohort of 269 participants, comprising 130 males and 149 females, selected without bias towards age, race, or ethnicity. 191 were confirmed as stroke-positive with diffusion-weighted MRI (DWI), and 78 were diagnosed with other conditions. Table I shows a demographic breakdown, summarizing key attributes relevant to stroke diagnosis and computer vision research.

### B. iOS Application Design

The M³Stroke model is deployed onto Apple iOS devices and is compatible with the latest iPads and iPhones. It is being tested using the iPad Pro M series. The choice of this device is motivated by its superior processing speed and expansive screen, which are expected to improve user experience, particularly for senior patients. The large screen aids in easier content reading and provides a clearer view of the "Cookie Theft" picture. Screenshots demonstrating the application's interface and features are shown in Fig. 1 (a).

[1]The research was collaboratively conducted by Houston Methodist Hospital, Texas, USA, and The Pennsylvania State University. All patients or legal guardians provided written informed consent. This study received approval from the Institutional Review Boards (IRBs) of both participating institutions: Houston Methodist IRB protocol No.PRO00020577 and Penn State IRB site No. SITE00000562. Patient recruitment was carried out at the Eddy Scurlock Stroke Center at Methodist. The anonymized data collected was then transmitted to Penn State for model development and testing.

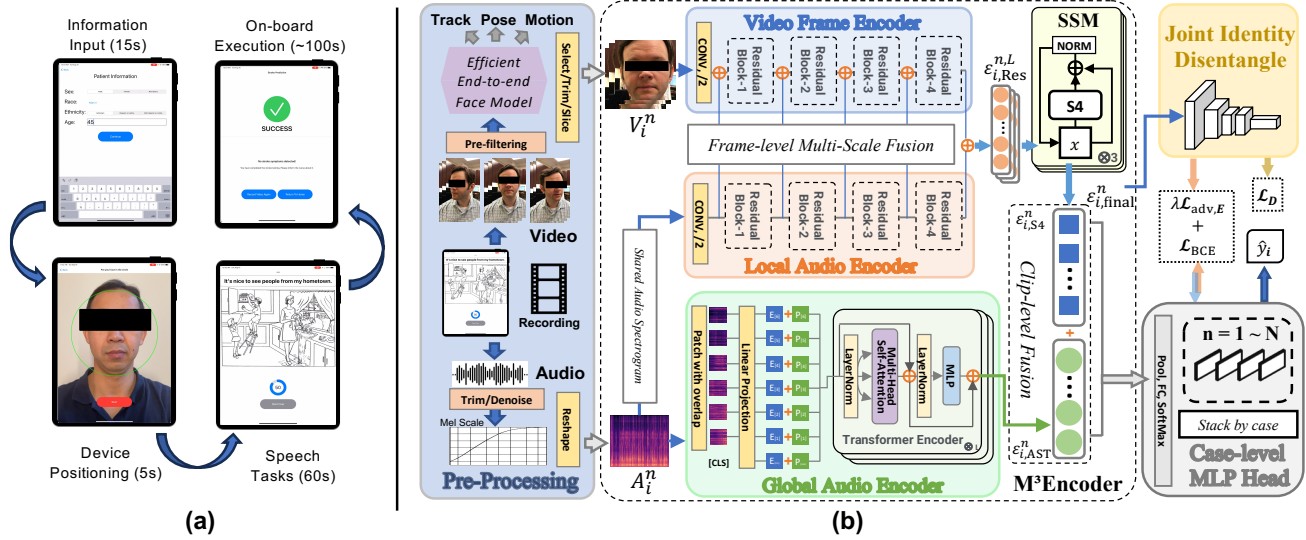

Fig. 1. Overview of the $\texttt{M}^3$Stroke framework. **(a)** Demonstration with screenshots from the iOS application. The images are screenshots captured from an Apple iPad Pro with an M1 chip. The step-by-step execution times are average durations, calculated based on the performance across the overall cohort. **(b)** Structure of the $\texttt{M}^3$Stroke model. Pre-processing results in two branches of data forwarded to the subsequent encoder modules, where $V_i^n$ and $A_i^n$ denote the video clip and audio segment for case $i$ clip $n$. $\bigoplus$ indicates the weighted addition of features, $+$ indicates the concatenation of feature vectors. $\texttt{[CLS]}$ is the added learnable class embedding. $\mathcal{E}_{i,\text{final}}^n$ is encoded feature of case $i$ clip $n$. $\mathcal{L}_{adv}$ is the training loss for the disentanglement network D and $\mathcal{L}_{\text{adv},E}$ is the imposed adversarial loss on the entire $\texttt{M}^3$Encoder.

TABLE I
A SUMMARY OF PARTICIPANTS' DEMOGRAPHIC INFORMATION.

| | Group | Stroke (n=191) | Non-Stroke (n=78) |
|---|---|---|---|
| **Sex** (%) | Male | 90 (47.12%) | 40 (51.28%) |
| | Female | 101 (52.88%) | 38 (48.72%) |
| **Ethnicity** (%) | Hispanic | 14 (7.33%) | 6 (7.69%) |
| | Non-Hispanic | 175 (91.62%) | 66 (84.62%) |
| | Opt-out | 2 (1.05%) | 6 (7.69%) |
| **Race** (%) | African American | 63 (32.98%) | 24 (30.77%) |
| | Other | 128 (67.02%) | 54 (69.23%) |

The development dataset underwent a preliminary filtering process to maintain data quality. The category 'Sex' refers to the physiological characteristics of the patients.

*1) Application workflow:* The iOS application is designed to efficiently record patient information and interactively guide users in optimally positioning the device for enhanced recording quality. Initially, the data is saved locally on the device to expedite processing. Subsequently, it is transmitted to external storage systems. The on-device execution consists of data processing, feature extraction, and model inference. Following these processes, a probability outcome is then provided to the user based on the screening. The application can achieve stroke screening for the most challenging non-obvious cases within a satisfactory execution time, as indicated in the figure (typically under three minutes per screening). The recording files are pre-processed before being fed forward to the encoder. Any recordings shorter than 30 seconds or lacking complete medical information are regarded as incomplete and excluded from further processing.

*2) Optimized face manipulations:* The video frames are processed with the Apple Vision framework's internal de-

tection, tracking, and motion estimation to crop out near-frontal face views with meaningful motions, instead of using separate models in prior work *DeepStroke* [12]. Specifically, the facial landmarks determine the square detection/tracking bounding box with the landmark's geometric center, and estimate the non-facial motion by a rolling sum of roll/pitch/yaw statistically, instead of performing optical-flow extraction. By optimizing the face manipulations, the pre-processing time is reduced by over 70% with improved stability and quality.

*3) Data processing:* The extracted frame segments are then compiled into video clips comprising $L$ frames each to minimize file I/Os. The corresponding audio is first extracted into Waveform Audio File Format (.wav) and equally sliced to match the clips. The sliced audio segments are transformed into log-mel spectrograms. We extracted $N$ fixed-length video clips and audio segments/spectrograms for each case.

### C. $\texttt{M}^3$Encoder

To facilitate accurate diagnosis and efficient execution of the iOS pipeline, we propose a novel **M**ulti-scale, **M**ulti-level, **M**ulti-modal fusion ($\texttt{M}^3$) Audio-Visual encoder structure that demonstrates stronger feature extraction for stroke-related patterns. The $\texttt{M}^3$ Encoder incorporates effective sequence feature extraction, efficient state-space temporal sequence modeling, and a novel feature aggregation scheme.

*1) Multimodal feature extraction:* Due to the heterogeneous nature of the pre-processed data, we first forward them to different feature extraction modules to generate meaningful high-dimensional embeddings. For case $i$, the $n^{\text{th}}$ video clip, denoted as $V_i^n = \{f_i^{n,1}, f_i^{n,2}, ..., f_i^{n,L}\}$, is processed frame-by-frame as a sequence of features using a ResNet-based embedding network (Video Frame Encoder in Fig. 1 (b)) pre-trained on the FairFace dataset [29]. We denote the generated

feature at the end of the $s^{\text{th}}(s \in [1,4])$ network block as $\mathcal{E}^{n,l,s}_{i,\text{Video}}$ for case $i$ and frame $l$. The FairFace pre-training aims to mitigate facial appearance biases commonly seen in the field of computer vision. The corresponding audio segment is processed into $M$ log-mel bins with hidden dimension $D$ (i.e., $A^n_i \in \mathbb{R}^{D \times M}$) and forwarded to an Audio Spectrogram Transformer (AST) [11] (Global Audio Encoder in Fig. 1 (b)) pre-trained on the large-scale AudioSet dataset [30] as embedding $\mathcal{E}^n_{i,\text{AST}}$. The audio transformer model aims to extract global audio information by leveraging the transformer's attention mechanism. To encourage better video-audio feature fusion, we feed the same spectrogram to a ResNet-based Local Audio Encoder to capture more fine-grained spectrum information with convolutional kernels. The Local Audio Encoder is a ResNet-18 pre-trained on the ESC-50 dataset [31]; we denote generated feature vectors as $\mathcal{E}^{n,l,s}_{i,\text{Audio}}$. Pre-trained audio models ensure the transfer of domain knowledge from the two comprehensive datasets, thereby minimizing the risk of overfitting when working with more limited clinical datasets.

*2) State-space temporal sequence modeling:* To model the temporal properties of stroke features from video and audio sequence features, two computational challenges need to be tackled. First, the video sequences are relatively longer compared to traditional action recognition tasks, and due to gradient problems, traditional Recurrent Neural Networks (RNNs) or Long Short-Term Memory (LSTM) methods are not suitable. Second, deploying the model on a mobile device requires the model to be lightweight, and the current transformers suffer from the quadratic complexity of self-attention mechanisms, making them too computationally heavy for the purpose. In response to these challenges, our approach harnesses the advanced capabilities of the latest state-space model (SSM) for effective sequence modeling. In SSM, a sequence of input $u(t)$ is mapped to a higher dimensional latent space $x(t)$ and then projected to the output $y(t)$, as illustrated in Eq. 1.

$$x'(t) = \mathbf{A}x(t) + \mathbf{B}u(t), \qquad y(t) = \mathbf{C}x(t) + \mathbf{D}u(t) \quad (1)$$

In this model, parameters $\mathbf{A}$, $\mathbf{B}$, $\mathbf{C}$, and $\mathbf{D}$ are learned by gradient descent. However, the conventional implementation of SSM is known to encounter gradient-related issues. The S4 model addresses this challenge by incorporating the High-Order Polynomial Projection Operator (HiPPO) theory of continuous-time memorization [32] to enable the state $x(t)$ to effectively 'memorize' the historical data of the input $u(t)$, enhancing the model's ability to capture and retain relevant information.

We have specifically opted for the Structured State Space sequence model (S4) [10] that achieves SoTA performance on a range of benchmarks with weaker inductive biases, while being computationally efficient. The S4 model utilizes an efficient parameterization of SSM by decomposed the HiPPO matrix as the sum of a normal and low-rank matrix. The operation allows the recurrence and convolution operation to be optimal or near-optimal. In the deep neural network setting, S4 uses a 1-D sequence map and make $H$ self copies to handle higher-dimensional features, analogous to depthwise-separable

convolutions in CNN. They are mixed with a position-wise linear layer and activated with nonlinear functions to allow overall deep SSM represent nonlinearity.

*3) Feature aggregation:* The aggregation of our audio and video representations follows a multi-scale, multi-level fusion. First, we perform a frame-level fusion and leverage stage-wise lateral fusion to perform multi-scale feature fusion of ResNets by injecting the ResNet-18 audio feature into the ResNet-34 video branch, thanks to the same interim feature dimensions of the two models. Specifically, we add the audio features after block $s$ to the corresponding video feature: $\mathcal{E}^{n,l,s'}_{i,\text{Video}} = \mathcal{E}^{n,l,s}_{i,\text{Video}} + \tau\mathcal{E}^{n,l,s}_{i,\text{Audio}}$, where $\tau$ denotes the tunable audio information injection rate. $\mathcal{E}^{n,l,s'}_{i,\text{Video}}$ is forwarded to the video network's block $s + 1$. This ensures both video and audio features are coded by the state-space module. We denote the generated multimodal feature after all ResNet blocks as $\mathcal{E}^{n,l}_{i,\text{Res}}$ for the case $i$, clip $n$, and frame $l$. These features are forwarded to the S4 module together for a clip-level embedding $\mathcal{E}^n_{i,\text{S4}} = S4(\{\mathcal{E}^{n,1}_{i,\text{Res}}, \mathcal{E}^{n,2}_{i,\text{Res}}, ..., \mathcal{E}^{n,L}_{i,\text{Res}}\})$. Second, we perform a clip-level fusion by concatenating and normalizing features generated from the AST with the output of the state-space module before applying fully connected layers as the final clip-level feature $\mathcal{E}^n_{i,\text{final}} = \text{Norm}(\mathcal{E}^n_{i,\text{S4}}, \mathcal{E}^n_{i,\text{AST}})$, following a late fusion scheme.

*4) Training losses:* The training of the model follows an adversarial training scheme. We denote the $\text{M}^3$ encoder as E and adopt an auxiliary discriminative network as the joint identity disentangle module. The identity disentangle module (denoted as D) follows two losses, adversarial and classification. The adversarial loss aims to guide the network to generate identity-free audio-visual features. D concatenates final feature pairs from clip $n_1$ of sample $i$ and clip $n_2$ of sample $j$ as $\text{D}(\mathcal{E}^{n_1}_{i,\text{final}}, \mathcal{E}^{n_2}_{j,\text{final}})$, using corresponding matching label $i == j$ indicating if the two samples are from the same patient. The training loss $\mathcal{L}_\text{D}$ for the joint identity disentangle module takes the form of Mean Squared Error (MSE):

$$\mathcal{L}_\text{D} = \sum_{i,n} \left\| i == j - \text{D}(\mathcal{E}^{n_1}_{i,\text{final}}, \mathcal{E}^{n_2}_{j,\text{final}}) \right\|_2 \quad (2)$$

The classification loss $\mathcal{L}_\text{cls}$ is composed of two distinct components. The first component is a clip-level binary cross-entropy loss, where we compare case label $y_i$ with the prediction to each clip, denoted as $\hat{y_i}^n = \text{Softmax}(FC(\mathcal{E}^n_{i,\text{final}}))$, and $w_i$ is the weight for case $i$. The second component is an adversarial loss imposed on the encoder. This loss is designed to adversarially promote uncertainty in the output of D, thereby enhancing the robustness of the model against overfitting and improving its ability to generalize to unseen data. The total training loss $\mathcal{L}_\text{E}$ is the weighted sum of $\mathcal{L}_\text{BCE}$, and $\mathcal{L}_\text{adv,E}$ with tunable $\lambda$

$$\mathcal{L}_\text{BCE} = -\sum_{i,n} w_i[y_i \log \hat{y_i}^n + (1 - y_i)(1 - \log \hat{y_i}^n)]$$

$$\mathcal{L}_\text{adv,E} = -\sum_{i,n} \left\| \frac{1}{2} - \text{D}(\mathcal{E}^{n_1}_{i,\text{final}}, \mathcal{E}^{n_2}_{j,\text{final}}) \right\|_2 \quad (3)$$

$$\mathcal{L}_\text{E} = \mathcal{L}_\text{BCE}, +\lambda\mathcal{L}_\text{adv,E}$$

The training for E and D is conducted iteratively. During each iteration, the training alternates between two steps: first, the parameters of E are frozen, and D is updated; subsequently, D is frozen, and E is updated.

## IV. IMPLEMENTATION AND RESULTS

### A. Implementation details

*1) Mobile deployment:* Our data collection protocol utilized the device's cameras and built-in recording function to capture videos coupled with audio recorded at a 44.1 KHz sampling rate using AAC-LC coding. The synchronized video and audio were automatically saved as H.264 MOV files, with MPEG-4 compression. The app allowed users to choose between the front and rear cameras, facilitating self-assessment by patients capable of holding the phone themselves. In scenarios where the front camera was used, the "cookie-theft" picture was digitally displayed on the device's screen. Regardless of the camera used, videos were consistently recorded at 1080p@30fps. When the patient is unable to steadily hold the device, the screening will be conducted by a clinician. A printed "cookie-theft" picture will be provided for the patients, who were positioned as standing, sitting, or lying in bed, and instructed to face the device.

Our iOS app is compatible with devices running iOS version 16.0 or higher. The app was developed using Xcode 14.3.1, under the SwiftUI framework and the CoreML SDK. We utilized the Apple Vision framework's API `VNDetectFaceLandmarksRequest`, to perform face detection, tracking, and the acquisition of facial landmarks for head pose estimation. To facilitate the loss-less conversion of the trained PyTorch model into a CoreML model, we followed a two-step process. First, we traced the PyTorch model using random tensor data. Second, we used the Unified Conversion API's `convert()` method of CoreML Tools to convert the traced model into a CorelML model and save it as a .mlmodel file, enabling direct instantiation within an iOS app. The input parameters of the generated CoreML model were represented as multi-dimensional arrays (MLMultiArray) and pre-loaded into the installed app. Due to Apple's incomplete implementation of certain complex64 type operators of PyTorch in the latest version of coremltools (7.1), we modified the source code of coremltools to ensure support for these operators, including `Torch.view_as_complex` and `Torch.conj`.

*2) Model implementation:* We perform the model training on a PC workstation using the Python PyTorch environment. We set the pre-processing to extract $N = 7$ video clips of length $L = 64$ using standard ResNet implementations and skipped the final decoders, making $\mathcal{E}_{i,\text{Res}}^{n,l} \in \mathbb{R}^{512}$. We set $M = 128$ audio log-mel bins and used a hidden dimension $D = 600$, giving $\mathcal{E}_{i,\text{AST}}^{n,l} \in \mathbb{R}^{768}$. We followed the implementation of the original Structure State-Space Model (S4) [33] as a PyTorch module to take extracted, flattened video and audio features. We set an internal dropout rate to 0.2 to prevent overfitting. The injection rate for audio features $\tau$ was set to 0.5. The output feature of $\texttt{M}^3$ encoder $\mathcal{E}_{i,\text{final}}^{n}$ was reshaped to a stack of $8 \times 8$ features to fit the FCN operations during adversarial learning. The identity-disentanglement module D adopts a small full convolutional network (FCN) with three convolution layers and outputs a vector of $\mathbb{R}^2$. Both ResNets and the AST backbone parameters are frozen while the S4 module, the final case-level MLP head, and the identity disentangle module have gradient updates. The encoder loss used $\lambda = 5$. We used a batch size of 32 for model training and an initial learning rate of 0.0001 that decays to 10% every 20 epochs. The best-performing model on the validation set was saved and used for holdout testing benchmark. The specificity, sensitivity, accuracy, and area under the ROC curve (AUC) are calculated and reported to compare the models and baselines.

### B. Model performance

*1) Prospective Hold-out Evaluation:* In prospective testing for stroke triage, the process is framed as a binary classification problem in machine learning: distinguishing stroke cases from non-stroke cases, which excludes the more detailed subtypes of stroke not typically covered in triage. Discharge diagnoses are used to define the binary ground truth for stroke or transient ischemic attack (TIA). A prospective testing scheme is adopted to benchmark the model performance by stratifying the data into chronological sets for training, validation, and testing. Specifically, data collected before Feb. 2021 (n=181 with 131 strokes & 50 non-strokes) are used for training, data from February 2021 to May 2021 (n=41 with 28 strokes & 13 non-strokes) for model validation, and data after May 2021 (n=47 with 32 strokes & 15 non-strokes) for testing. The distribution of strokes and non-strokes follows the real-world scenario during stratification and avoids biases of estimation. Training involves random shuffling of the training set, with validation after each epoch to retain the best model while keeping the test set unseen. During inference, the models yield a real-valued probability of stroke between 0 and 1. The **A**rea **U**nder the Receiver Operating Characteristic **C**urve (AUC) is used as the performance metric since it examines every cutoff threshold in [0,1] and is independent of the potential class imbalance. For each model/baseline, we also examine the specificity and sensitivity to compare with the pre-hospital triage performance reported by the stroke center of the hospital over years on patients coming into the ER with suspicion of a stroke. We report the test performance metrics against triage staff, the *DeepStroke* model [12], and the cutting-edge face video representation model, MARLIN [13], showcasing the comparative results in Table II.

The $\texttt{M}^3$Stroke model (AUC=0.7229) demonstrates a significant performance gain over baselines. $\texttt{M}^3$Stroke achieved as high as 80.85% accuracy, 60.00% specificity, and 90.63% sensitivity on the test set, attaining 16.82% accuracy gain, 14.29% specificity gain, and 20.44% sensitivity gain over the triage performance. When contrasted to its predecessor, the *DeepStroke* model (AUC=0.6542), $\texttt{M}^3$Stroke achieved a 17.02% accuracy gain, 13.33% specificity gain, and 18.75% sensitivity gain. The SoTA face video analysis model MARLIN (AUC=0.6063) is a pre-trained Video Masked Autoencoder (VideoMAE) with top performance on various face-

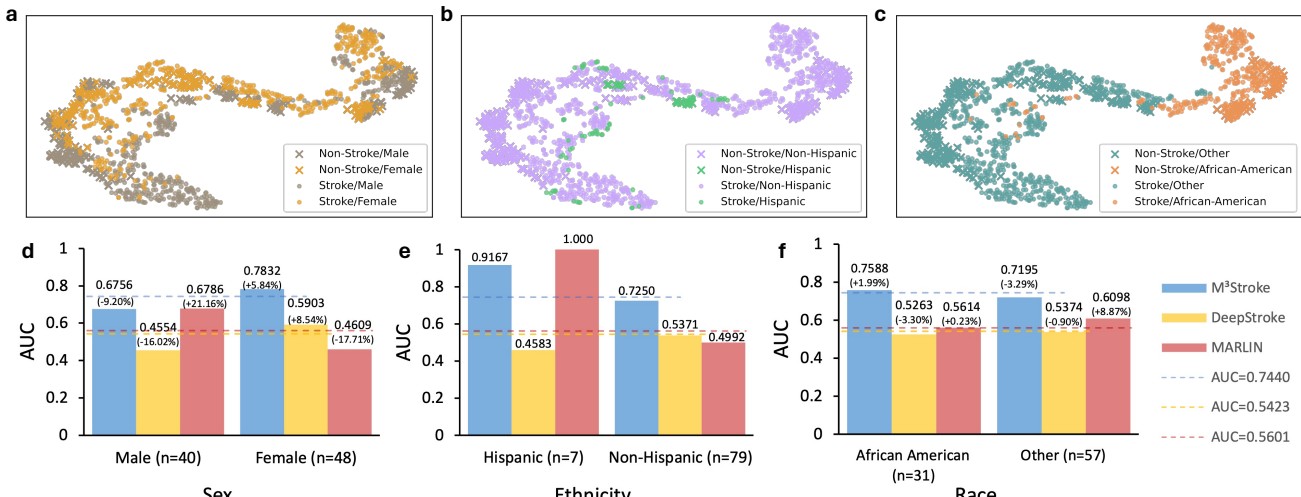

Fig. 2. Visualizations for model fairness of the M³Stroke model and baselines. **a-c**, t-SNE plots for sex, ethnicity, and race, respectively, on generated features from the training set, where the model's predictions perfectly aligned with the ground truth. **d-f**, bar plots and AUC lines to compare AUC change for the M³Stroke model and baselines. Due to an insufficient Hispanic cohort, the percentage changes are not calculated.

TABLE II
RESULTS OF THE PROSPECTIVE HOLDOUT TESTING (N=47).

| | Accuracy | Specificity | Sensitivity | AUC |
|---|---|---|---|---|
| Pre-Hospital Triage | 64.03 | 45.71 | 70.19 | - |
| M³Stroke | **80.85** | **60.00** | **90.63** | **0.7229** |
| *DeepStroke* | 63.83 | 46.67 | 71.88 | 0.6542 |
| MARLIN | 65.96 | 40.00 | 78.13 | 0.6063 |

The pre-hospital triage performance represents a cumulative statistic from the emergency center over years on patients coming into the ER with suspicion of a stroke. AUC values are not calculated for triage due to binary decisions. The tunable cutoff thresholds are determined by the validation set on a best-performing epoch to maximize the geometric mean while maintaining a minimum of 40% specificity or 70% sensitivity.

related benchmarking datasets. Despite this, its performance did not surpass traditional triage methods, suggesting that the screening of minor strokes might require the identification of unique features typically not prioritized in general facial analysis tasks. The design choices and performance gain are detailed in the ablation study section.

*2) Model Fairness Evaluations:* We present our effort in addressing major fairness concerns regarding different attribute groups. Despite not using patient demographics in the development of our model, we recognize the potential for a deep learning model to inadvertently capture stroke-irrelevant features and learn correlations based on them. Therefore, we examined sex (male/female), ethnicity (Hispanic/non-Hispanic), and race (African American/other) which have long been significant in both the field of computer vision and stroke diagnosis. We compare the final encoded features over different demographic groups using t-Distributed Stochastic Neighbor Embedding (t-SNE) on the training data. The results are shown in Fig. 2.

We notice that the encoded feature for different ethnicity groups is overall very mixed, suggesting that these differences are not captured by the model. Meanwhile, features associated with sex and race appear more distinct and separable in the

t-SNE plots, albeit the predictions (marked by a dot or cross) are independent of the existence of these clear clusters. These t-SNE plots suggest the overall prediction, or the decision boundary of the modal, is independent of these demographic variables when generating the probability of stroke, indicating that the model succeeds in achieving demographic disentanglement and fairness preservation.

We also report the AUC on all unseen data (encompassing both validation and test sets) and their percentage deviation from the overall performance, as well as a group of bar plots for clear visualization. The results are shown in Fig. 2 **d-f**. All attribute groups demonstrated relatively low error rates with the M³Stroke method. Compared with the SoTA face representation, the M³Stroke model shows enhanced performance stability and fairness. This shows the value of *FairFace* pre-training in generating facial representations that are more independent of appearance-based features, and can substantially reduce the race and ethnicity biases that are prevalent in the field of computer vision.

*C. Ablation Studies*

To examine the effectiveness of our proposed method, we ablated the M³Stroke model components and examined the performance gains with the addition of these modules. This involved retraining the model under various configurations while maintaining consistency in the training/validation protocols and optimizer settings. Our ablation study includes: "No ADV", the model without adversarial training; "No Pre-Train", the model that uses the default ImageNet pre-training; "No ADV", the model without AST audio branch; "S4-Video", the multimodal sequence encoding model with S4 representation; "AST Only", the stand-alone AST audio model.

The elimination of the adversarial training (M³Stroke vs. No ADV, and No AST vs. S4-Video) resulted in a drop in the validation and test performance, indicating the importance of identity disentanglement. Using generic pertaining showed a

TABLE III
ABLATION STUDY RESULTS ON BOTH VALIDATION AND TEST SETS.

| | Model Components | | | | AUC | |
|---|---|---|---|---|---|---|
| | ADV | AST | S4 | Pre-Train | Val | Test |
| M³Stroke | ✓ | ✓ | ✓ | ✓ | **0.7885** | **0.7229** |
| No ADV | ✗ | ✓ | ✓ | ✓ | 0.6868 | 0.6000 |
| No Pre-Train | ✓ | ✓ | ✓ | ✗ | 0.7060 | 0.5541 |
| No AST | ✓ | ✗ | ✓ | ✓ | 0.7637 | 0.6958 |
| S4-Video | ✗ | ✗ | ✓ | ✓ | 0.7527 | 0.6000 |
| AST Only | ✗ | ✓ | ✗ | ✓ | 0.6758 | 0.5958 |

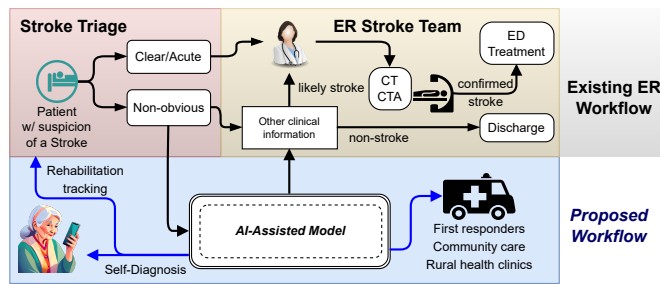

Fig. 3. The anticipated clinical and ER integration of M³Stroke.

clear performance drop from M³Stroke, showing the value of domain-specific transfer learning when medical training data is limited. The dual-branched multimodal sequence encoding model with S4 representation (No AST) can be seen as a variant of the *DeepStroke* model with S4 temporal modeling, which achieves relatively good performance. The stand-alone AST model does not directly show good diagnostic results, but provides an extra performance boost to the dual-branch encoder, by leveraging different audio understanding.

## V. DISCUSSION

In this work, we propose M³Stroke, a powerful and efficient mobile tool for mild to moderate stroke triage that aims to tackle the challenges of data scarcity, noise complexity, and feature subtlety. It is capable of resolving the long-standing data scarcity issue with its lightweight and user-friendly mobile application. The mobile collection protocol can be easily adapted to other conditions including but not limited to Alzheimer's disease, Parkinson's disease, etc, to effectively produce comprehensive, high-quality datasets. M³Stroke is a significant step forward from systems that are built on lab-collected, noise-free data. The face manipulations remove non-frontal and non-facial movements; the dual audio embedding captures multi-level information to mitigate noises; the identity disentanglement reduces the demographic biases, jointly achieving stroke triage "in the wild". The proposed M³Stroke model also bridges the research gap in audio-visual multimodal AI for efficient long sequence modeling with state-space modeling and the multi-scale multi-level feature fusion, specifically designed to identify more subtle facial movements and speech audio patterns for a comprehensive analysis of minor stroke symptoms.

M³Stroke is designed to be seamlessly integrated into various clinical stages, as depicted in Fig. 3. It facilitates self-evaluation for patients who suspect a stroke but lack overt symptoms. In conventional pre-hospital stroke screening, individuals often rely on emergency services or their own judgment when experiencing minor symptoms. A majority of missed stroke patients at the pre-hospital stage are due to un-awareness of the serious consequences and reluctance to seek medical attention. Our tool leverages mobile computing to provide accurate, efficient, and accessible guidance directly to patients at the scene. Second, it serves as an essential resource

for healthcare providers, particularly in settings where experienced neurologists are scarce or unavailable, such as rural communities. By equipping general healthcare providers with our tool, the accuracy in identifying minor strokes improves significantly, enabling timely interventions without specialized neurology or cardiovascular training. Third, there is potential for M³Stroke to be adapted to support stroke rehabilitation. In the later stage of the stroke recovery, symptoms of neurological deficits can become milder and harder to recognize. It can be used to monitor the progression of a patient's recovery, tracking subtle changes in stroke probability and aiding in the reduction of radiation exposure from neuroimaging.

We suggest several avenues for future research in this area:

- *Generalizing framework to different stroke severity:* The current development of M³Stroke specifically aims at mild to moderate stroke scenarios, but we anticipate the same method to generalize to other levels of severity. Further research can collect similar data and train another model to differentiate between normal and clear strokes. A rule-based speech/face test protocol can be used as the first-stage trigger that switches between minor/severe modes.
- *Refining pre-trained models with medical data:* Our method employs transfer learning with general-purpose pre-trained models, which may not optimally capture patient-specific features. Future research could focus on customizing these pre-trained models with medical data, thereby enhancing their accuracy and relevance in clinical settings.
- *Adapting the model for multilingual environments:* Our current model has limited testing in multilingual contexts, particularly for Hispanic populations. This aspect of research could involve developing and testing models trained on diverse linguistic datasets, ensuring the tool's accessibility and accuracy across different language-speaking communities.

## VI. CONCLUSION

We introduced M³Stroke, a powerful and efficient mobile tool for stroke triage, particularly effective in identifying mild to moderate cases with subtle symptoms, to provide new insights and exemplar pathways to the non-contact, natural camera-based computer-aided diagnosis of neurological disorders. M³Stroke aims to advance clinical screening of mild to moderate stroke that is prone to misdiagnoses, and substantially improve the stroke patient's quality of life. As

the first to achieve unified multi-media data collection and AI-based patient screening on a mobile device, $M^3$Stroke benefits from a novel audio-visual multimodal sequence encoder with confirmed superior robustness, representational capabilities, and fairness across diverse demographics with comprehensive experiments, underscoring its clinical applicability.

Overall, the integration of multimodal mobile AI into emergency triage marks a significant progression in medical technology, poised to revolutionize emergency medicine. By harnessing the capabilities of various data types, including video and audio, and melding these with mobile technology, these AI systems can be customized to identify and rank a variety of critical health situations beyond strokes, encompassing trauma injuries, heart attacks, and acute allergic reactions. The ability of these systems to swiftly and accurately process and analyze disparate data streams from mobile devices in real-time empowers first responders and medical professionals to make quicker, more informed decisions.

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
