# OpenReview forum: "$\texttt{M}^3$Stroke: $\textbf{M}$ulti$\textbf{M}$odal $\textbf{M}$obile AI for Emergency Triage of Mild to Moderate Acute Strokes"
_IEEE.org/EMBS/BHI/2024/Conference — IEEE BHI'24_

### Official Review · Reviewer_zjBd · 2024-07-30

**Overall Rating:** 7
**Confidence:** 5

**Other Quality Metrics:**

Clarity of writing: Excellent

Clinical Significance: Excellent

Methodological Novelty: Great

Experiments and Results: Great

**Questions For The Authors:**

1)	Please describe in more detail the differences between your study and [14].
2)	The authors use a highly imbalanced dataset. Please clarify how this dataset has been split into a train, validation, and test set. How many subjects per category (stroke, non-stroke) are included in the train, validation, and test set?
3)	Are there any prior works? Related work section is missing. You could also cite more research works in Introduction.
4)	Please, improve Fig. 1.
5)	In Abstract, please mention the limitations of the existing studies and how the proposed study addresses these limitations.

**Strengths:**

1)	Well written and organized.
2)	Novel enough for publication.
3)	A lot of experiments conducted.

**Summary Of The Paper:**

This paper presents a mobile tool based on multimodal learning for stroke triage.

**Weaknesses:**

1)	Highly imbalanced dataset.
2)	Related work section is missing.
3)	The abstract does not highlight the limitations of the existing studies and the novelties of the proposed approach.
4)	Figure 1 is a bit confusing. For instance, it is not obvious that the spectrogram is used as input both to the local audio encoder and to the global audio transformer.

---

### Official Review · Reviewer_SeEj · 2024-07-31
**The authors developed a system for the detection of mild-to-moderate acute strokes cases for emergency triage**

**Overall Rating:** 8
**Confidence:** 4

**Other Quality Metrics:**

(a) Clarity of writing; Good
(b) Clinical Significance; Great
(c) Methodological Novelty; Great
(d) Experiments and Results; Good

**Questions For The Authors:**

-	269 patients seem quite insufficient as the amount of data to be collected for a solution so complex. How can the authors prove there is no overfitting?
-	The authors limited the study to people which did not have pronounced stroke-related impairments. Can the authors speculate on how the system would work in subjects at high risk instead?
-	It is unclear how the severity of the stroke conditions were assessed
-	DWI abbreviation is not explained
-	Figures are not in order of appearance
-	Fig 1 is not well described. I’d prefer a more step-by-step visualization especially of the pre-processing or the SSS steps.
-	Why the authors adopted accuracy/specificity/sensitivity as metrics of choice, leaving out other metrics typically used for imbalanced datasets (AUC-PR, F1-score, etc.)?
-	I think the Discussion section is too long at this point
-	The performance reported may not be sufficient for real-world adoption; can the authors comment on this?

**Strengths:**

This paper is an interesting work

**Summary Of The Paper:**

The authors developed a system for the detection of mild-to-moderate acute strokes cases for emergency triage

**Weaknesses:**

This paper requires some modifications to be accepted (see below)

---

### Official Review · Reviewer_BXQ4 · 2024-08-11
**Paper 56**

**Overall Rating:** 7
**Confidence:** 3

**Other Quality Metrics:**

Clarity of writing: Excellent
Clinical significance: Great
Methodological novelty: Great
Experiments and results: Excellent

**Questions For The Authors:**

Is IRB (Institutional Review Board) obtained for the study? If yes, I suggest to include IRB details in data collection.
Were users having any difficulties in using the mobile app especially when the recording is more than 30 seconds? Perhaps some users’ experience can be included in the discussion.

**Strengths:**

This is a very well-written paper and thorough study. Within the page limit, the paper explains the key details of their methodology and evaluation. They also provide fair comparison and evaluation with other tools and studies. In addition, the paper highlights the demographic fairness in analysis to address major fairness concerns regarding different attribute groups. This is very commendable and shows the thoroughness in evaluation. The ablation study helps to demonstrate why ADV and AST combined can boost the performance and therefore substantiates why multimodal information is useful and works.

**Summary Of The Paper:**

This paper presents the integration of multimodal AI (audio and video) into emergency triage to advance clinical screening of mild to moderate stroke. The paper reported the performance of the multimodal AI model on a dataset of 269 patients and obtains 14% gain in specificity and over 20% higher sensitivity compared with traditional stroke triage methods. The paper also reported comparison results with existing tools such as DeepStroke, MARLIN. Their multimodal AI application shows great potential to advance hospital triage for stroke patients.

**Weaknesses:**

I was concerned on majority of a particular race among participants in Table 1 because it may bring bias to the study. However, later authors address this concern in their analysis.

---

### Decision · Program_Chairs · 2024-09-23

Accept